# Quantum Information Entropy of Hyperbolic Potentials in Fractional Schrödinger Equation

**DOI:** 10.3390/e24111516

**Published:** 2022-10-24

**Authors:** R. Santana-Carrillo, Jesus S. González-Flores, Emilio Magaña-Espinal, Luis F. Quezada, Guo-Hua Sun, Shi-Hai Dong

**Affiliations:** 1Centro de Investigación en Computación, Instituto Politécnico Nacional, UPALM, Ciudad de Mexico 07738, Mexico; 2Research Center for Quantum Physics, Huzhou University, Huzhou 313000, China

**Keywords:** hyperbolic potential well, fractional Schrödinger equation, Shannon entropy, Fisher entropy, 03. 67. -w, 03. 67.-a

## Abstract

In this work we have studied the Shannon information entropy for two hyperbolic single-well potentials in the fractional Schrödinger equation (the fractional derivative number (0<n≤2) by calculating position and momentum entropy. We find that the wave function will move towards the origin as the fractional derivative number *n* decreases and the position entropy density becomes more severely localized in more fractional system, i.e., for smaller values of *n*, but the momentum probability density becomes more delocalized. And then we study the Beckner Bialynicki-Birula–Mycieslki (BBM) inequality and notice that the Shannon entropies still satisfy this inequality for different depth *u* even though this inequality decreases (or increases) gradually as the depth *u* of the hyperbolic potential $U_{1}$ (or $U_{2}$) increases. Finally, we also carry out the Fisher entropy and observe that the Fisher entropy increases as the depth *u* of the potential wells increases, while the fractional derivative number *n* decreases.

## 1. Introduction

Quantum information entropy has been studied widely since Shannon proposed the classical concept in 1948 [1,2,3,4,5,6,7,8,9,10,11,12,13,14,15,16,17,18,19,20,21,22,23,24,25,26,27,28,29,30,31]. This is because the Shannon entropy as a measure of uncertainty is a generalization to the traditional Heisenberg relation. It should be recognized that all the contributions to this topic are concerned with the traditional Schrödinger equation, i.e., the fractional derivative number *n* is taken as an integer 2. That is to say, this research has been widely applied in many fields such as Harper model [27], complex quasi-periodic potential models [28] and the nanoelectronic dispositives [32]. Since the pioneering work on the fractional Schrödinger equation by the Laskin and others [30,33,34,35], however, this topic has attracted attention to many authors [36,37,38,39,40,41,42,43,44,45]. This is because the fractional Schrödinger equation can exhibit some quantum effects on quantum quantities due to the fractional derivative number *n*. For instance, the typical applications of fractional calculus in quantum mechanics include the energy band structures [36], the position-dependent mass fractional Schrödinger equation [37], the fractional harmonic oscillator [38], the nuclear dynamics of molecular ion ${H_{2}}^{+}$ [39], the propagation dynamics of light beams [40], the spatial soliton propagation [41], the Rabi oscillations [42], the self focusing and wave collapse [43] and others.

As mentioned above, the Shannon information entropy has been studied for many quantum soluble potentials. Based on their analytical wave functions, it is not difficult to calculate the wave function in the momentum space by studying the traditional Fourier transformation. Among these soluble potentials, the hyperbolic potentials have played an important role in semiconductor physics, e.g., the development of related research on graphene [14,46,47,48,49]. Nevertheless, a number of quantum systems cannot be solved analytically except for numerical study. Stimulated by our previous work [30,45], we are going to study the quantum information entropy for hyperbolic potentials $U_{1,2}$ in the time-independent fractional Schrödinger equation. The hyperbolic potential wells, which belong to single-well potentials with a minimum value *u*, are defined as [29,50]
(1)$$U_{q}\left(x;u\right)=\{\begin{array}{cc} -\frac{\hbar^{2}}{2M}\frac{u}{\cosh^{4}\left(x\right)}, & q=1,\\ \frac{\hbar^{2}}{2M}\left(-u+\frac{u\sinh^{4}\left(x\right)}{\cosh^{4}\left(x\right)}\right), & q=2. \end{array}$$

Up to now, due to the computational complexity of quantum information of the fractional Schrödinger equation including the calculation of the wave function and the Fourier transform, almost all authors have paid attention to the quantum information of the traditional Schrödinger equation, i.e., the derivative number n=2 except for our previous study [45], in which n∈(1,2] was taken. In that work [45], we have carried out the quantum information entropies of multiple quantum well systems in fractional Schrödinger equations. Recently, we have studied the Shannon entropy of the traditional (n=2) Schrödinger equation [29] with these two potentials (Equation 1) as shown in Figure 1, but we do not show how the fractional derivative *n* affects the wave function and the relevant physical quantities such as the position and momentum probability density as well as the Shannon entropy. Moreover, in the present study we will display their behaviours through taking the noninteger n∈(0,2], which is different from our previous study n∈(1,2] taken in Ref. [45]. Certainly, we will see that the calculation results would become not ideal when *n* is taken too small. On the other hand, in order to compare with the global characteristic of Shannon entropy, it is necessary to study the locality of Fisher entropy.

This work is organized as follows. In Section 2 we present the formalism to solve a fractional Schrödinger equation system in numerical way for hyperbolic potentials ($U_{1,2}$). In Section 3 we present the obtained results including the wave functions of hyperbolic potential wells, the position and momentum entropy densities and the position $S_{x}$ and momentum $S_{p}$ Shannon entropies by varying the depth *u* of the potential wells. Furthermore, we show the characteristics of the 11th excited state case except for the ground, first, second and sixth states. In particular, we show how the fractional derivative number *n* affects their physical features. Finally, to compare with the Shannon entropy, we also study the Fisher entropy. In Section 4 we summarize our conclusions.

## 2. Formalism

Let us start with the one dimensional fractional Schrödinger equation
(2)$$\left(-\frac{\hbar^{2}}{2M}\frac{\partial^{n}}{{\partial |x|}^{n}}+U_{q}\left(x;u\right)\right)\varphi \left(x\right)=E\varphi \left(x\right),$$
where the fractional derivative with a noninteger number *n* is defined as follows:(3)$$\frac{\partial^{n}\varphi \left(x\right)}{{\partial |x|}^{n}}=\frac{1}{2\cos\left(\frac{n\pi }{2}\right)\Gamma \left(2-n\right)}\frac{d^{2}}{dx^{2}}\int_{-\infty }^{\infty }{\left|x-\xi \right|}^{1-n}\varphi \left(\xi \right)d\xi .$$
The *E* and $\varphi \left(x\right)$ represent the eigenvalues and the eigenfunctions, respectively.

To solve this Equation (Equation 2), we have to use a numerical method to calculate the fractional derivative. Let us define
(4)$$g_{k}=\frac{{\left(-1\right)}^{k}\Gamma \left(n+1\right)}{\Gamma \left(\frac{n}{2}-k+1\right)\Gamma \left(\frac{n}{2}+k+1\right)},$$
where k=0,1,2,⋯n>0, then one has
(5)$$g_{0}\geq 0,\;\;g_{-k}=g_{k}\leq 0,\;\;\left|k\right|\geq 1.$$
Now, the fractional centered difference is defined by
(6)$$\Delta_{h}^{n}f\left(x\right)=\sum_{k=-\infty }^{\infty }g_{k}f\left(x-kh\right).$$
In this way, we have
(7)$$-\frac{1}{h^{n}}\Delta_{h}^{n}f\left(x\right)=\frac{\partial^{n}}{\partial |x{|}^{n}}f\left(x\right)+O\left(h^{2}\right).$$
When *h* tends to 0, then $\frac{\partial^{n}}{\partial |x{|}^{n}}f\left(x\right)$ becomes the fractional derivative for 0<n≤2. Now, we can rewrite Equation (Equation 2) as,
(8)$$\sum_{j=0}^{N}A_{ij}\varphi_{j}=E\varphi_{j}.$$ This matrix eigenvalue problem can be diagonalized by the available routine libraries [51].

After solving this equation, the eigenfunction can be obtained by this way. As shown in Figure 2, we notice that the wave function will move towards the origin as the fractional derivative number *n* decreases and the crest of the wave functions rises up gradually. This makes the wave function localized near the origin.

Once the wave function is obtained, then the Shannon information entropy densities $\rho_{s}\left(x\right)$ and $\rho_{p}\left(p\right)$ can be calculated as,
(9)$$\begin{array}{c} \rho_{s}\left(x\right)={\left|\varphi \left(x\right)\right|}^{2}\ln\left(|\varphi \left(x\right){|}^{2}\right),\\ \rho_{s}\left(p\right)={\left|\varphi \left(p\right)\right|}^{2}\ln\left(|\varphi \left(p\right){|}^{2}\right), \end{array}$$
where the wave function in the momentum space $\varphi \left(p\right)$ can be calculated by the Fourier transformation
(10)$$\varphi \left(p\right)=\frac{1}{\sqrt{2\pi }}\int \varphi \left(x\right)e^{-ipx}dx.$$ For the present study, however, we have to use the fast Fourier method to obtain the integrals numerically.

Based on them (Equation 9), the Shannon information entropies for position $S_{x}$ and momentum $S_{p}$ can be calculated respectively by
(11)$$\begin{array}{c} S_{x}=-\int_{-\infty }^{\infty }\rho_{s}\left(x\right)dx,\\ S_{p}=-\int_{-\infty }^{\infty }\rho_{s}\left(p\right)dp. \end{array}$$

For the sum of the position and momentum entropy $S_{x}$ and $S_{p}$, Beckner, Bialynicki-Birula and Mycieslky found an important uncertainly relation [52,53]
(12)$$S_{x}+S_{p}\geq D\left(1+\ln\pi \right)$$
where *D* stands for the spatial dimension (D=1 here). This inequality implies that if the momentum entropy $S_{p}$ increases then the position entropy $S_{x}$ will decrease and vice versa.

Before ending this section, let us consider the Fisher information. Its importance was noticed by Sears et al. [54], who found that the kinetic energy could be considered as a measure of the information distribution. To know its wide applications, the reader can refer to our recent study [55] for more information. It should be pointed out that the local character of Fisher entropy is a main difference compared with Shannon information. The Fisher entropy is defined as an expectation value of the logarithmic gradient of density or as the gradient functional of density, i.e., its explicit definition is given by [56]
(13)$$I_{F}=\int_{a}^{b}\frac{{\left[\rho^{\prime }\left(x\right)\right]}^{2}}{\rho (x)}dx=4\int_{a}^{b}{\left[\psi^{\prime }\left(x\right)\right]}^{2}dx$$
where the probability density is defined as $\rho \left(x\right)={\left|\varphi \left(x\right)\right|}^{2}$.

## 3. Results and Discussions

We are now in the position to present the results of this work. Motivated by our recent study about Shannon entropy in multiple well quantum well system in fractional Schrödinger equations [45], we present the normalized wave function, the position and momentum entropy densities $\rho_{x}$ and $\rho_{p}$ as well as the Shannon entropy. For simplicity, we show the results for the ground, the first, second, sixth excited states and much higher excited 11-th state. As shown in Figure 3 and Figure 4, we show the variation of the position and momentum entropy density as functions of the variables *x* and *p*, respectively. The ground, the first, second and sixth excited states are denoted by solid blue line, red dotted line, green dashed line and black dash-dotted line, respectively. Likewise, the values of noninteger *n* are taken as 2,1.3,0.8,0.3, respectively. It is found that the position entropy density ρ(x) becomes more localized but the momentum entropy density ρ(p) is more delocalized as the noninteger *n* decreases. For example, the crest of the wave function for the ground state decreases gradually but those for the excited states increase gradually with the decreasing *n*. However, for the case of momentum entropy density as shown in Figure 4, the crest of the wave function for the ground state decreases gradually as the *n* decreases.

In addition, we study the position and momentum entropy densities for higher excited states, say 11-th excited state, as shown in Figure 5 and Figure 6. It is found that the crest of position entropy density gradually becomes larger as the noninteger *n* decreases, but it is contrary to the momentum entropy density except that the position and momentum entropy density becomes more localized and delocalized, respectively.

Now, we explore the Beckner Bialynicki-Birula–Mycieslki inequality. This inequality is still satisfied for both potential $U_{1,2}$. This can be verified by Figure 7, Figure 8, Figure 9 and Figure 10. Figure 7 and Figure 9 show the position and momentum entropy $S_{x}$ and $S_{p}$ as a function of the depth *u* for the potential wells $U_{1,2}$. For simplicity, we consider the ground state for two potentials. We notice that the position entropy $S_{x}$ decreases with increasing depth *u* of the potential well but decreasing noninteger *n*, but the momentum entropy $S_{p}$ is contrary to that of $S_{x}$ case. In Figure 8 we observe the sum $S_{x}+S_{p}$ for $U_{1}$ decreases with the depth *u* of the potential well, while that of case $U_{2}$ increases with depth *u*. This implies that the sum $S_{x}+S_{p}$ for $U_{1}$ will become more stable than that of case $U_{2}$ when the depth *u* increases. Certainly, their sum always remains above the minimum value of the value 2.144. On the other hand, we notice from Figure 8 and Figure 10 that the difference of the sums for various *n* is very small and the sum is always around 3.15.

Finally, let us show the plots of the Fisher entropy as a function of the depth *u* of the potentials. As shown in Figure 11, we observe that the Fisher entropy increases with the increasing *u* but with the decreasing *n* for both potentials $U_{1,2}$. The Fisher entropy $I_{F}$ for $U_{1}$ is greater than that of the $U_{2}$. This can be explained well by the shape of the potentials since the potential $U_{1}$ is narrower than $U_{2}$.

## 4. Concluding Remarks

In this work we have carried out a class of hyperbolic potentials within the frame of time-independent fractional Schrödinger equation and verified that the BBM inequality is still satisfied for the different fractional derivative number *n*. We have shown how this number *n* affects the wave function, the entropy density, Shannon entropy in the position and momentum space. We have noticed that the position and momentum entropy density becomes more localized and delocalized, respectively as the noninteger *n* decreases. On the other hand, we have noted that the $S_{x}$ decreases as the depth *u* of the potential wells increases and the noninteger *n* decreases, but it is contrary to the momentum $S_{p}$ case. Finally, we have shown that the sum of entropy $S_{x}+S_{p}$ for $U_{1}$ will become more stable than that of case $U_{2}$ when the depth *u* increases. This is determined by the shape of the potential wells as shown in Figure 1. This is because potential well for $U_{1}$ becomes narrower than that of $U_{2}$. The motion of particles in potential well $U_{1}$ is more restricted than that in potential well $U_{2}$. Finally, we have also carried out the Fisher entropy and observed that the Fisher entropy increases as the depth *u* of the potential wells increases and the *n* decreases.

## Figures and Tables

**Figure 1 entropy-24-01516-f001:**
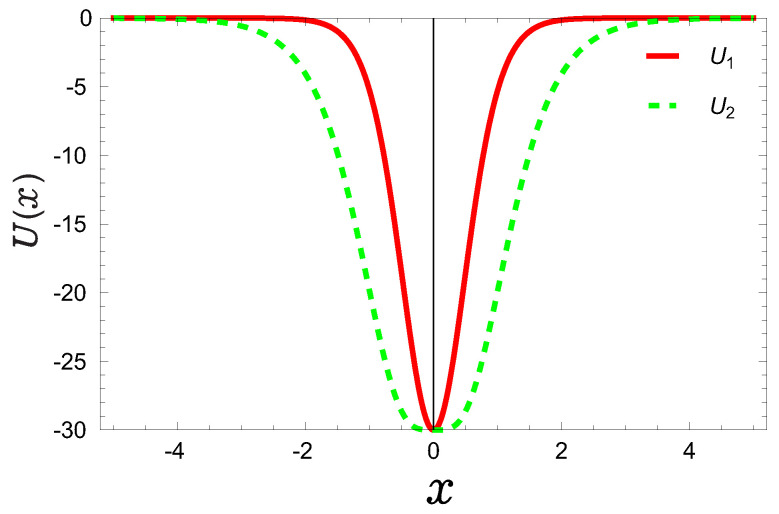
Plot of single quantum wells $U_{1,2}$ (multiplied by a unit $2M/\hbar^{2}$) as a function of variable *x* with a depth of potential well u=30.

**Figure 2 entropy-24-01516-f002:**
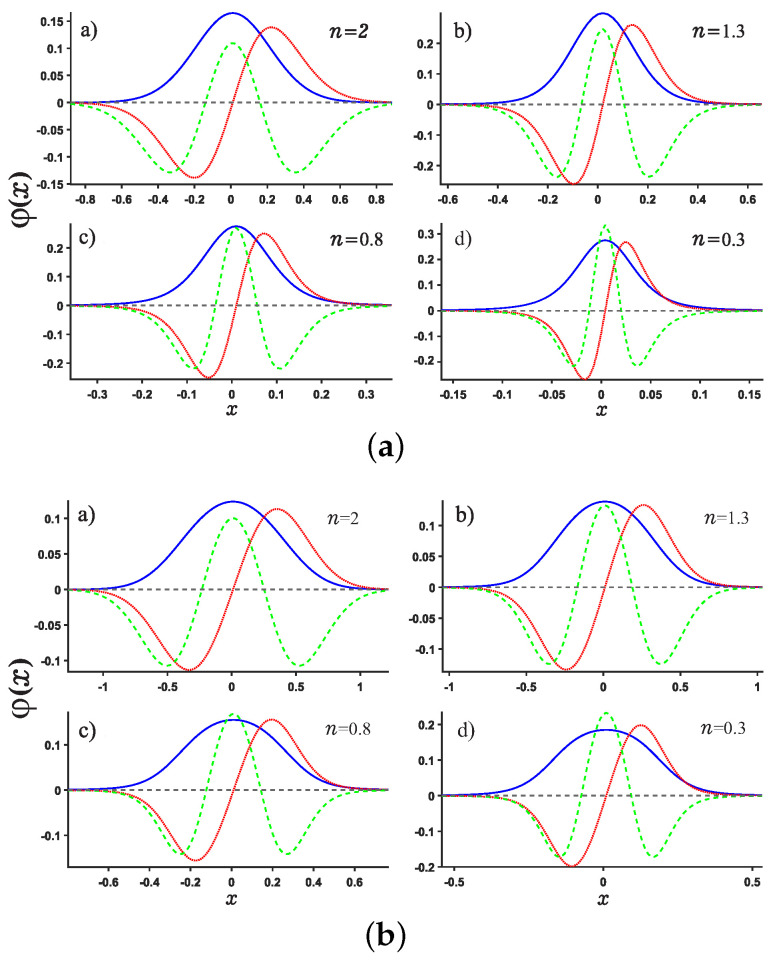
The normalized wave functions as a function of position *x* for single-well potentials $U_{1}$ (see Panel (**a**)) and $U_{3}$ (Panel (**b**)), respectively. The solid blue line, red dotted line and green dashed line denote the ground state, the 1st and 2nd excited states, respectively. The noninteger *n* is taken as n=2,1.3,0.8,0.3, respectively.

**Figure 3 entropy-24-01516-f003:**
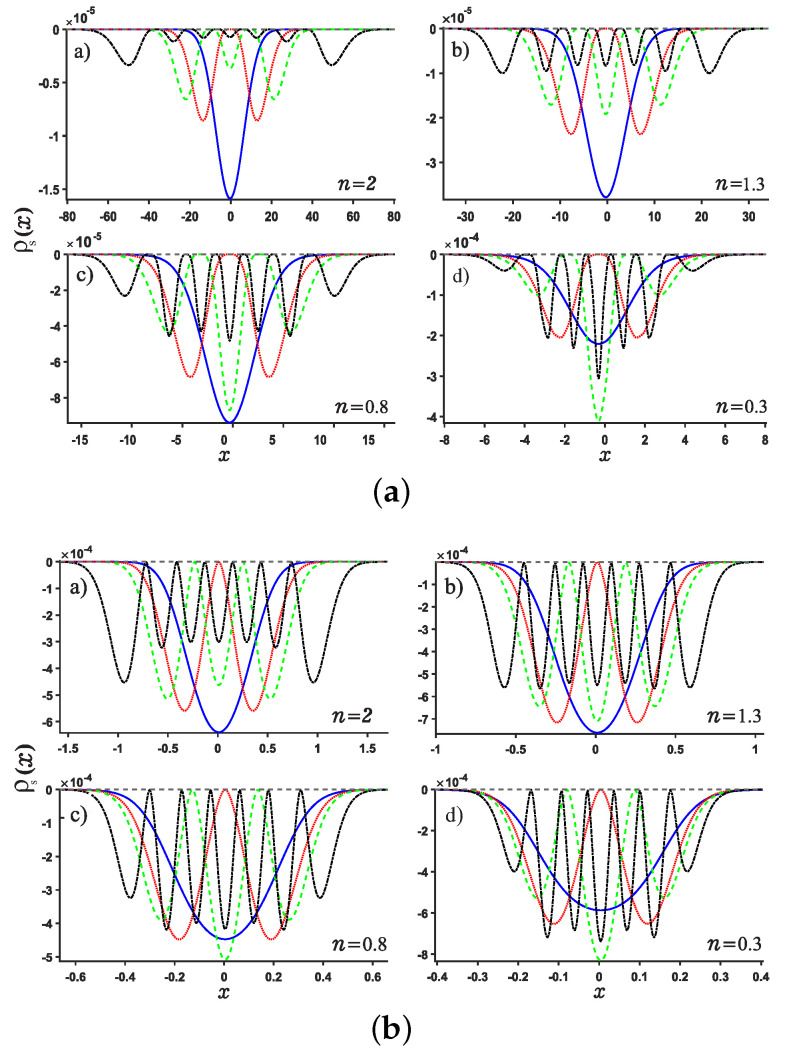
Plots of the position entropy density as a function of the position *x* for potential wells $U_{1}$ (Panel (**a**)) and $U_{2}$ (Panel (**b**)). For each panel, Panels (**a**–**d**) are plotted for different values of noninteger *n*, say n=2,1.3,0.8,0.3, respectively. The ground state, the first, second and sixth excited states are denoted by solid blue line, red dotted line, green dashed line and black dash-dotted line, respectively.

**Figure 4 entropy-24-01516-f004:**
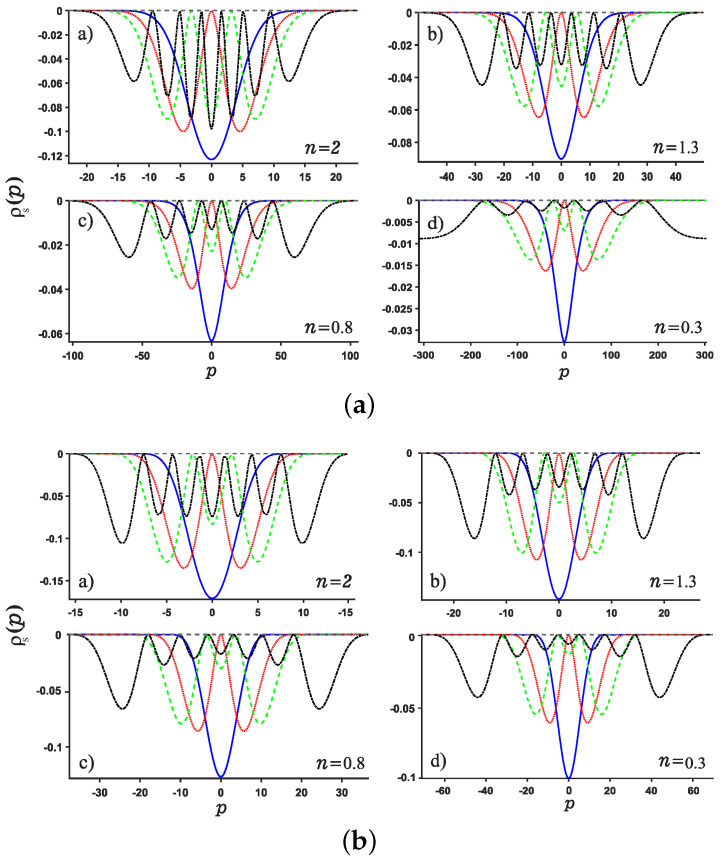
Same as above Figure 3 but for the momentum *p* case for the potentials $U_{1,2}$.

**Figure 5 entropy-24-01516-f005:**
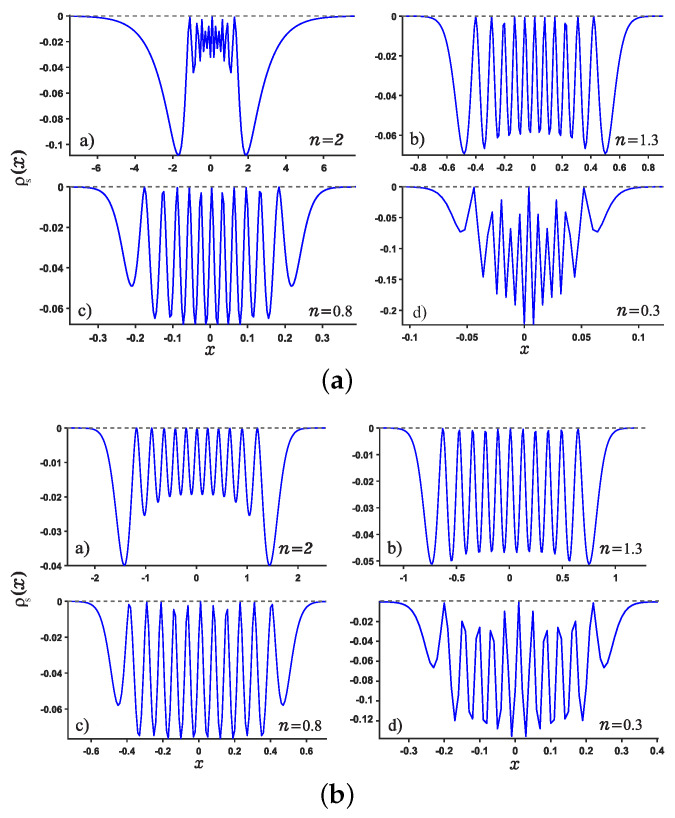
Plots of position entropy density for normalized 11-th excited state for potential wells $U_{1}$ (see Panel (**a**)) and $U_{2}$ (see Panel (**b**)). For each Panel, Panels (**a**–**d**) are plotted for different values of *n*.

**Figure 6 entropy-24-01516-f006:**
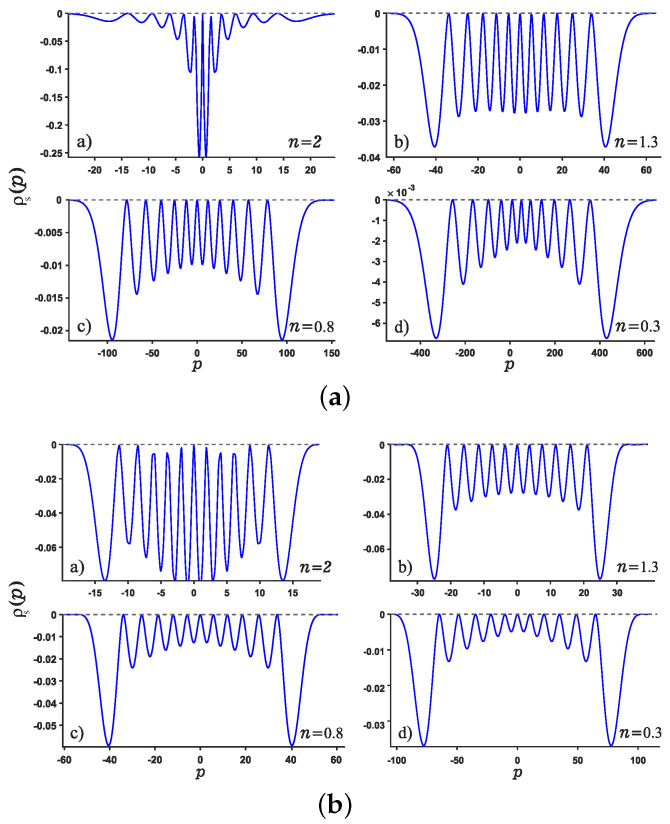
Same as Figure 5 but for the momentum entropy density.

**Figure 7 entropy-24-01516-f007:**
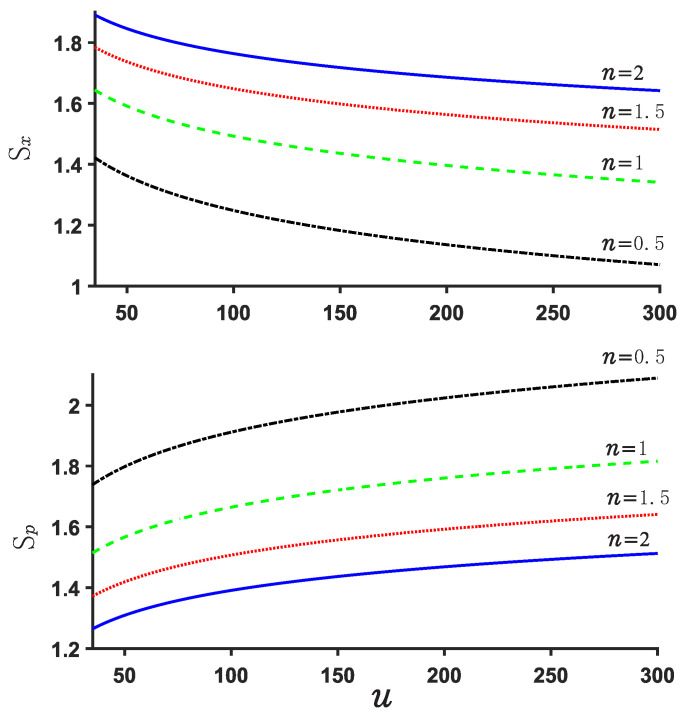
$U_{1}$ position and momentum entropy $S_{x}$ and $S_{p}$ for different *n*. We consider the ground state here.

**Figure 8 entropy-24-01516-f008:**
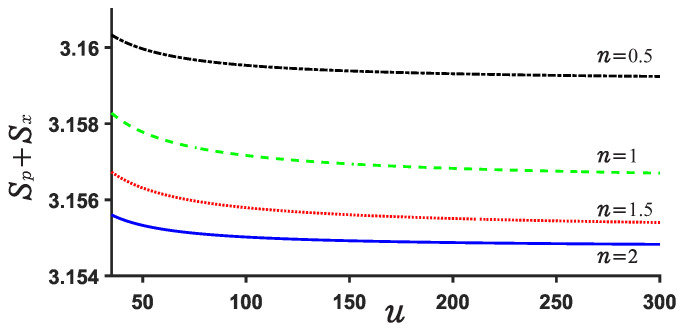
Sum of position and momentum entropy $S_{x}+S_{p}$ for different values of *n* and *u* in the case $U_{1}$. The ground state is considered.

**Figure 9 entropy-24-01516-f009:**
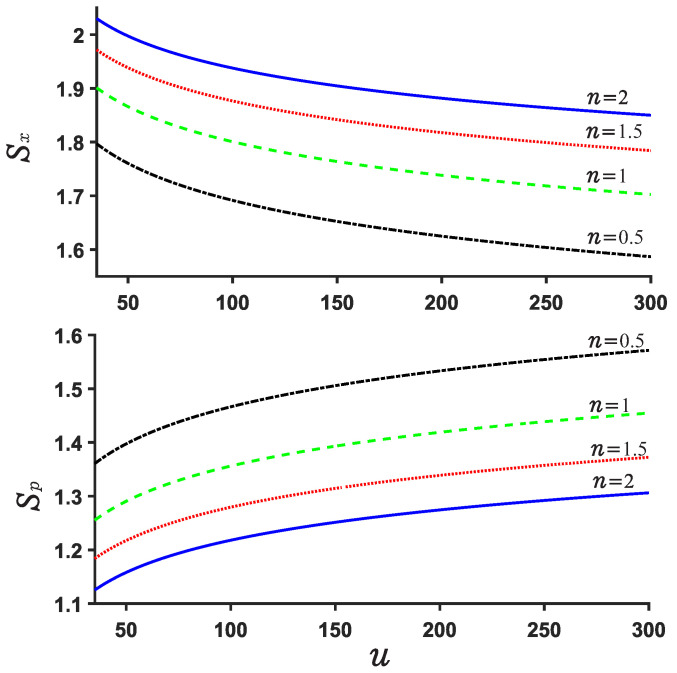
Position and momentum entropy $S_{x}$ and $S_{p}$ for various *n* in the case $U_{2}$.

**Figure 10 entropy-24-01516-f010:**
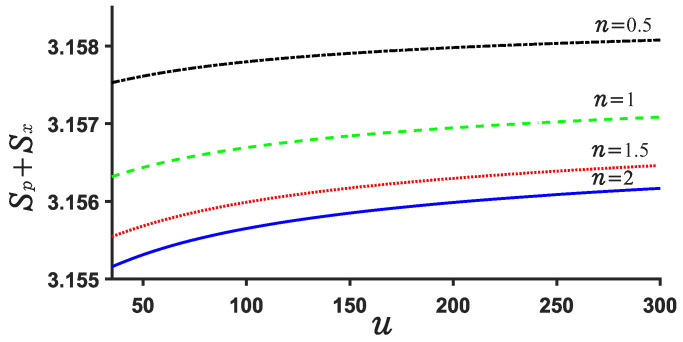
Sum of position and momentum entropy $S_{x}+S_{p}$ for various *n* and *u* for $U_{2}$.

**Figure 11 entropy-24-01516-f011:**
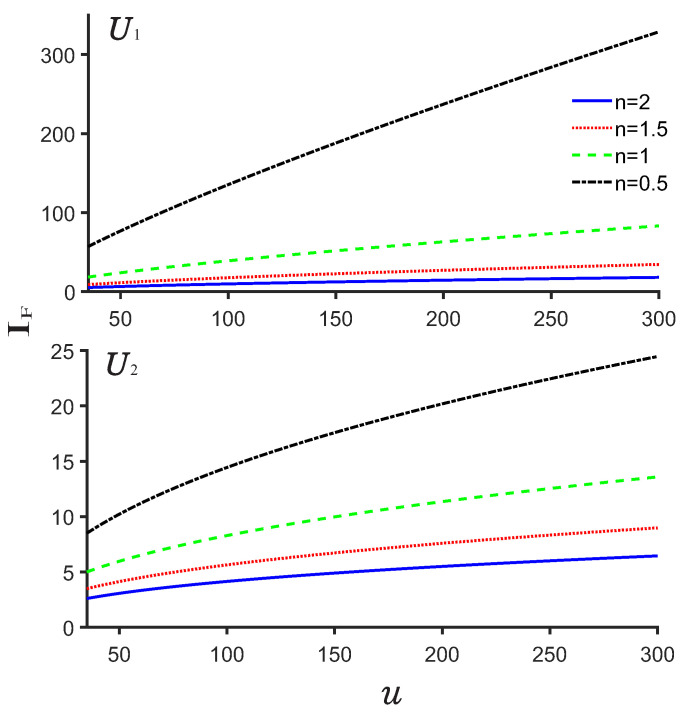
Plots of the Fisher entropy as a function of the depth *u* for the potentials $U_{1}$ and $U_{2}$.

## Data Availability

Not applicable.

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
