# Peer review of "Quantum Information Entropy of Hyperbolic Potentials in Fractional Schrödinger Equation"

_entropy, 2022, doi:10.3390/e24111516_

Round 1

Reviewer 1 Report

I find the manuscript on the information entropy of hyperbolic potentials in fractional Schroedinger equation interesting, well-written and well-founded scientifically. The authors detail a mainly numerical study of information theoretic objects in this non-traditional version of the Schroedinger equation, which is nontheless well-known, and which has a well-recognized placed in the realm of physics models, albeit not a central one. The authors go through a series of calculations which have  the potential of being useful to researchers interested in questions of quantum information as they are related to fractional derivative models. I recommend this paper for publication in Entropy. 

Author Response

Dear Referee,

Many thanks for your positive report on this work.

Best regards

Shihai

Reviewer 2 Report

This work presents calculation of different quantum entropies based on solutions of the fractional Schroedinger equation. Unfortunately, neither the motivation nor the results are sound enough to warrant publication in Entropy. It seems to me that such problem could serve as a good exercise for graduate students but it does not meet the criteria for scientific publication. Therefore, I cannot recommend the publication.

Author Response

see separated file.

Reviewer 3 Report

Dear Editor, n this work, the authors studied the Shannon information entropy for two hyperbolic single-well potentials in the fractional Schr\"{o}dinger equation. The results are new and interesting and also considering the fractional Schr\"{o}dinger equation can be considered as a deformed quantum mechanics that are able to cover some unexpected results in physics. I think the paper is publishable in its current form and I would like to congratulate the authors because of the expansion of the edges of the science. Best Wishes Hassan Hassanabadi

Author Response

Dear Referee,

Many thanks for your positive report on our work.

Best regards

Shihai